# TopAdapter: Topology-Aware Prompt Tuning for Efficient Point Cloud Understanding

**Changshuo Wang** [1]   **Shuting He** [2]   **Xiang Fang** [3]   **Weijun Li** [4]   **Yixian Shen** [5]   **Mingkun Xu** [6]   **Zhongtian Sun** [7]   **Prayag Tiwari** [8]

## Abstract

Point cloud data, with its inherent geometric and topological structures, plays a critical role in 3D vision tasks. However, existing parameter-efficient fine-tuning (PEFT) methods predominantly focus on input token prompting, overlooking the intrinsic geometric information. To address this limitation, we propose **TopAdapter**, a novel PEFT framework that enhances geometric perception by injecting local topological information into pre-trained 3D vision models. **TopAdapter** leverages 0D, 1D, and 2D simplices from algebraic topology as fundamental building blocks, introducing two core modules: the Topology Injection module (ToInjection) and the Topology Transfer module (ToTransfer). ToInjection constructs multi-scale topological features using a simplex generator and dynamically fuses them with semantic features via a geometric controller, thereby enhancing geometric adaptability. ToTransfer propagates these topological primitives across Transformer layers, ensuring efficient transmission of geometric information. Extensive experiments demonstrate that **TopAdapter** outperforms existing PEFT methods, achieving performance comparable to full fine-tuning across various benchmarks.

---

[1]Department of Computer Science, University College London, London, United Kingdom [2]School of Computing and Artificial Intelligence, Shanghai University of Finance and Economics, Shanghai, China [3]ERI@N, Interdisciplinary Graduate Programme, Nanyang Technological University, Singapore [4]Institute of Semiconductors, Chinese Academy of Sciences, Beijing, China [5]Informatics Institute, University of Amsterdam, Amsterdam, The Netherlands [6]Guangdong Institute of Intelligence Science and Technology, Zhuhai, China [7]School of Computing, University of Kent, Canterbury, United Kingdom [8]School of Information Technology, Halmstad University, Sweden. Correspondence to: Xiang Fang <xfang95@gmail.com>.

*Proceedings of the 43rd International Conference on Machine Learning*, Seoul, South Korea. PMLR 306, 2026. Copyright 2026 by the author(s).

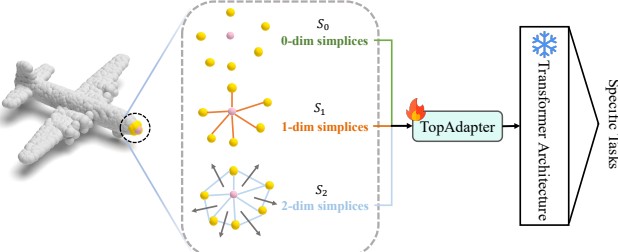

*Figure 1.* Overview of **TopAdapter**. Given a point (highlighted in pink) within an airplane point cloud, we explicitly characterize its local geometry by constructing simplices of varying dimensions within its neighborhood. These multi-dimensional simplices are concatenated to dynamically fine-tune Transformer-based architectures. By integrating geometric and topological information, **TopAdapter** achieves high expressiveness while maintaining an optimal balance between parameter efficiency and performance.

## 1. Introduction

The proliferation of advanced 3D scanning technologies (Wang et al., 2022) has established point cloud data as a fundamental representation in computer vision, enabling transformative applications across autonomous driving (Zhao et al., 2023; Chib & Singh, 2023), robotic navigation (Soori et al., 2023; Goel & Gupta, 2020), and 3D reconstruction (Wu et al., 2026; Sereno et al., 2020). Pre-trained 3D vision models (Chen et al., 2023; Qi et al., 2023) have demonstrated remarkable feature extraction capabilities. However, full fine-tuning demands substantial computational resources while risking catastrophic forgetting of valuable pre-trained knowledge.

Parameter-efficient fine-tuning (PEFT) (Jia et al., 2022) has emerged as a compelling solution by selectively updating a minimal subset of parameters. While PEFT methods (Lester et al., 2021; Shi & Lipani, 2024) have achieved success in 2D vision and NLP, point cloud data—with its irregular distribution, sparsity, and complex geometric structures—demands specialized treatment that existing methods fail to adequately address. Recent PEFT methodologies (Tang et al., 2024; Ai et al., 2025; Liang et al., 2025; Zhang et al., 2025a) for point clouds have made notable progress. IDPT (Zha et al., 2023) enhances instance aware-

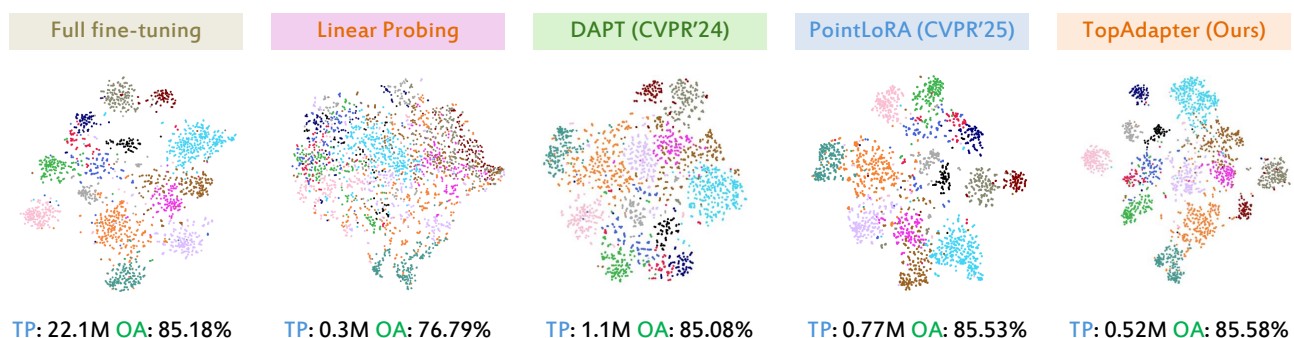

*Figure 2.* t-SNE visualization of feature distributions on the PB-T50-RS variant of ScanObjectNN (Uy et al., 2019) using different fine-tuning approaches with Point-MAE as the baseline model. TP: Tunable parameters. OA: Overall accuracy.

ness through dynamic prompts; DAPT (Zhou et al., 2024) employs low-rank decomposition; PointGST (Liang et al., 2025) optimizes spectral encoding; and GAPrompt (Ai et al., 2025) introduces geometry-aware prompts. However, these approaches consistently overlook intrinsic topological properties—surface connectivity, hole distributions, and geometric invariance. As illustrated in Fig. 2, this oversight limits model expressiveness under challenging conditions such as noise or occlusion.

To address these limitations, we propose **TopAdapter**, a novel PEFT framework that integrates topological principles with point cloud learning. As shown in Fig. 1, we incorporate simplices—fundamental geometric units from algebraic topology (Smith et al., 2021; Hensel et al., 2021)—as building blocks: zero-dimensional simplices (points) capture spatial positions, one-dimensional simplices (edges) encode connectivity, and two-dimensional simplices (faces) represent local surfaces. **TopAdapter** comprises two core modules: ToInjection constructs multi-dimensional simplices and dynamically integrates geometric features with semantic representations via a geometric controller; ToTransfer propagates topological information across model layers for effective geometric utilization. Theoretically, we demonstrate that **TopAdapter** can approximate point cloud topologies with arbitrary precision while maintaining robustness against perturbations.

Our principal contributions are:

1. We introduce **TopAdapter**, a novel parameter-efficient fine-tuning framework for point clouds that leverages simplices as fundamental geometric units to effectively capture topological and geometric structures.

2. We develop the Topology Injection and Topology Transfer modules to facilitate efficient extraction and propagation of geometric information, substantially enhancing the geometric perception capabilities of 3D vision models.

3. We provide rigorous theoretical foundations grounded in algebraic topology and the universal approximation theorem, establishing the mathematical validity and robustness of our approach.

4. Through comprehensive experimental evaluation, we demonstrate that **TopAdapter** significantly outperforms existing PEFT methods, establishing a promising new direction for point cloud processing research.

## 2. Related Work

### 2.1. Point Cloud Representation Learning

Deep learning has achieved remarkable progress in point cloud feature extraction. PointNet (Qi et al., 2017a) pioneered direct processing of raw point cloud data, addressing the fundamental challenge of permutation invariance through its innovative combination of per-point feature extraction and global aggregation. Building upon this foundation, PointNet++ (Qi et al., 2017b) introduced a hierarchical feature learning architecture that significantly enhanced local structure capture. Subsequent efforts, exemplified by DGCNN (Wang et al., 2019) and DAF-Net (Wang et al., 2025a), have continued to advance representation learning methodologies.

Inspired by the success of self-supervised pre-training in image domains, researchers (Pang et al., 2022; Zhang et al., 2022; Chen et al., 2023) have increasingly adapted these techniques to 3D point cloud processing. Current self-supervised approaches primarily fall into two categories: masked modeling and contrastive learning. Point-BERT (Yu et al., 2022) and Point-MAE (Pang et al., 2022) successfully transferred masked autoencoder concepts to 3D, developing rich representations by reconstructing masked regions. PointGPT (Chen et al., 2023) implemented an autoregressive generation framework analogous to GPT models, while ReCon (Qi et al., 2023) established a unified framework integrating both generative and contrastive learning paradigms.

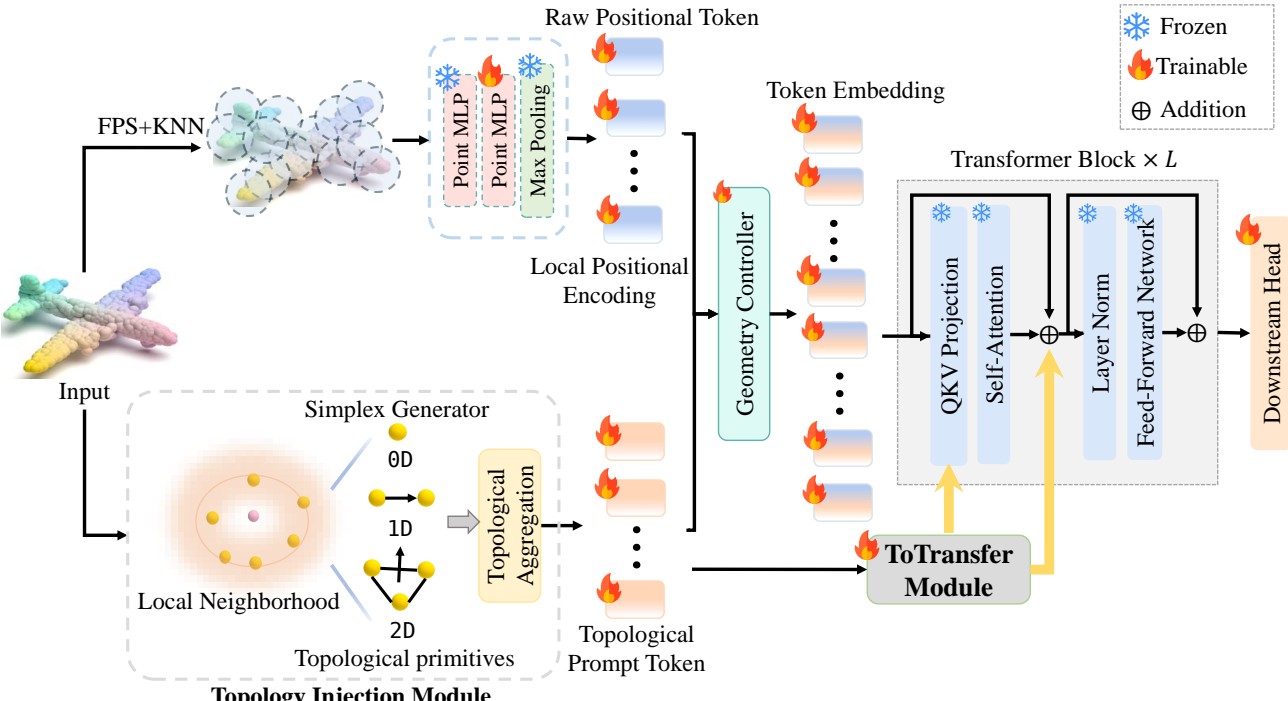

*Figure 3.* Overall architecture of **TopAdapter**. The framework comprises two main modules: (1) Topology Injection module (ToInjection), which constructs topological primitives from multi-dimensional simplices (0D, 1D, 2D) and integrates them via a geometry controller; and (2) Topology Transfer module (ToTransfer), which propagates topological information across Transformer layers. This design enables efficient capture and utilization of geometric and topological properties while maintaining minimal trainable parameters.

ACT (Dong et al., 2022) advanced cross-modal geometric understanding by leveraging established 2D and language foundation models.

### 2.2. Point Cloud Parameter-Efficient Fine-Tuning

Parameter-efficient fine-tuning (PEFT) (Lester et al., 2021; Shi & Lipani, 2024) aims to optimize knowledge transfer from pre-trained models to downstream tasks by selectively updating a minimal subset of parameters. In 2D vision and natural language processing, PEFT methodologies (Fu et al., 2023; Hu et al., 2022; He et al., 2023) are generally categorized into three principal approaches: prompt tuning, adapter tuning, and low-rank adaptation. However, research on PEFT specifically for point clouds remains in its early stages.

Current 3D PEFT approaches (Tang et al., 2024; Ai et al., 2025; Liang et al., 2025; Zhang et al., 2025a) have explored various strategies. IDPT (Zha et al., 2023) employs Edge-Conv networks to capture local token interactions and dynamically generate prompts. DAPT (Zhou et al., 2024) extends traditional adapter architectures by simultaneously producing scaling factors and prompts through dynamic adapters. Point-PEFT (Tang et al., 2024) enhances transfer learning by integrating prompt tuning, adapter tuning, and bias adjustment techniques. PPT (Zhang et al., 2025a)

leverages positional encoding to optimize prompt tuning, effectively capturing spatial distribution characteristics. Point-GST (Liang et al., 2025) investigates PEFT from a spectral perspective, adjusting features in the graph domain to better represent local geometric properties. PointLoRA (Wang et al., 2025b) combines low-rank adaptation with multi-scale token selection to capture both global and local features. GAPrompt (Ai et al., 2025) introduces geometry-aware prompts, enhancing geometric perception through specialized point prompts, point offset prompters, and prompt propagation mechanisms.

Despite these advances, existing methods generally overlook crucial aspects of point cloud representation: the intrinsic topological structures and effective propagation of geometric information across model layers. Our proposed **TopAdapter** addresses these limitations by introducing topological simplex primitives and cross-layer propagation mechanisms, establishing a novel perspective for geometric representation learning.

## 3. TopAdapter

### 3.1. Problem Definition

Parameter-efficient fine-tuning (PEFT) (Lester et al., 2021; Shi & Lipani, 2024) for point clouds aims to adapt pre-

trained 3D vision models (Zhang et al., 2025b; Liang et al., 2024) to downstream tasks by optimizing a minimal parameter subset while maintaining or enhancing model performance. Formally, given a pre-trained 3D vision model $f_\theta$, where $\theta$ represents the complete parameter set, the objective of PEFT is to identify a parameter subset $\phi$ such that $|\phi| \ll |\theta|$, while achieving performance comparable to or exceeding that of the fully fine-tuned model $f_{\theta'}$.

In the point cloud domain, the model input consists of a set of three-dimensional points $\mathcal{P} = \{\mathbf{p}_i \mid i = 1, \ldots, n\}$, where each point $\mathbf{p}_i \in \mathbb{R}^3$ represents spatial coordinates. Pre-trained models typically employ Transformer-based architectures that process point cloud data through multi-head self-attention mechanisms and feedforward networks.

### 3.2. Simplex Theoretical Foundation

A simplex (Hensel et al., 2021) represents a fundamental geometric concept, defined as the simplest polytope of a given dimension—the convex hull of $k + 1$ affinely independent points. Formally:

**Definition 3.1** (Simplex). Given a point set $\{\mathbf{v}_0, \mathbf{v}_1, \ldots, \mathbf{v}_k\} \subset \mathbb{R}^n$, a $k$-dimensional simplex $\sigma^k$ is defined as:

$$\sigma^k = \left\{ \sum_{i=0}^{k} \lambda_i \mathbf{v}_i \;\middle|\; \begin{array}{l} \lambda_i \geq 0 \quad \forall i \in \{0, \ldots, k\} \\ \sum_{i=0}^{k} \lambda_i = 1 \end{array} \right\}, \quad (1)$$

where $\lambda_i$ are barycentric coordinates.

Simplices are categorized by dimension: a 0D simplex (vertex) is a single point; a 1D simplex (edge) is a line segment connecting two points; a 2D simplex (face) is a triangle formed by three non-collinear points; and a 3D simplex (tetrahedron) is a solid formed by four non-coplanar points.

Simplicial homology, a branch of algebraic topology, enables systematic study of geometric properties through simplicial complexes and homology groups, characterizing features such as connected components, holes, and voids.

**Definition 3.2** (Simplicial Complex). A simplicial complex $\mathcal{K}$ is a collection of simplices satisfying:

1. If $\sigma \in \mathcal{K}$, then every face of $\sigma$ also belongs to $\mathcal{K}$.

2. The intersection of any two simplices in $\mathcal{K}$ is either empty or a common face.

Drawing upon topological theory, **TopAdapter** leverages simplices of varying dimensions as fundamental topological primitives.

### 3.3. Topology Injection Module

The Topology Injection module (ToInjection) constitutes the core component of **TopAdapter**, responsible for extracting geometric and topological features from point cloud data and integrating them into the model. Grounded in simplex theory, this module constructs an efficient mechanism to capture multi-scale geometric structures. As illustrated in Fig. 3, ToInjection comprises two principal components: the Simplex Generator and the Geometry Controller.

#### 3.3.1. SIMPLEX GENERATOR

The Simplex Generator constructs and processes simplices of different dimensions, enabling comprehensive representation of point cloud geometric structures. We establish the theoretical foundation through the following theorem:

**Theorem 3.3** (Topological Representation of Point Clouds). *Given a point cloud* $\mathcal{P} = \{\mathbf{p}_i \in \mathbb{R}^3\}_{i=1}^{N}$, *the simplicial complex* $\mathcal{K}_r$ *constructed through local neighborhoods can accurately represent the topological structure of* $\mathcal{P}$. *Within* **TopAdapter**, *the topological features extracted using 0D, 1D, and 2D simplices can effectively approximate the homology groups of* $\mathcal{K}_r$, *thereby capturing the geometric invariants of the point cloud.*

Based on this theorem, we construct local neighborhood structures through strategic sampling. We select $G$ centroids $\mathbf{c}_g \in \mathbb{R}^{G \times 3}$ using farthest point sampling (FPS) to ensure uniform coverage. For each centroid, we apply K-nearest neighbors (KNN) to construct local neighborhoods, yielding neighboring points $\mathbf{n}_{g,k} \in \mathbb{R}^{G \times K \times 3}$, where $K$ denotes the number of neighbors. We construct three types of simplicial features:

**0-dimensional simplices** represent positional information of individual points. The feature vector incorporates both centroids and their neighbors:

$$\mathbf{S}_0 = [\mathbf{c}_g, \mathbf{n}_{g,k}] \in \mathbb{R}^{G \times K \times 6}, \quad (2)$$

preserving absolute spatial information within local neighborhoods.

**1-dimensional simplices** represent edges connecting centroids to neighbors, capturing connectivity:

$$\mathbf{S}_1 = \mathbf{n}_{g,k} - \mathbf{c}_g \in \mathbb{R}^{G \times K \times 3}, \quad (3)$$

encoding relative spatial relationships and local geometric structure through displacement vectors.

**2-dimensional simplices** represent triangular faces formed by a centroid $\mathbf{c}_g$ and two neighbors $\mathbf{n}_{g,k1}$ and $\mathbf{n}_{g,k2}$, capturing surface properties such as curvature and orientation. We utilize normal vectors, angles, and triangle areas as representative features. For each centroid-neighbor pair, we identify

the nearest second neighbor to form a 2D simplex. The simplex edges are $\mathbf{e}_{g,k1} = \mathbf{n}_{g,k1} - \mathbf{c}_g$ and $\mathbf{e}_{g,k2} = \mathbf{n}_{g,k2} - \mathbf{c}_g$. Features are computed as:

$$\mathbf{normal} = \frac{\mathbf{e}_{g,k1} \times \mathbf{e}_{g,k2}}{\|\mathbf{e}_{g,k1} \times \mathbf{e}_{g,k2}\|}, \quad (4)$$

where $\times$ denotes cross product and $\|\cdot\|$ the Euclidean norm, yielding a unit normal vector.

$$\mathbf{A} = \frac{\|\mathbf{e}_{g,k1} \times \mathbf{e}_{g,k2}\|}{2}, \quad (5)$$

computing triangle area as half the cross product magnitude.

$$\theta = \arccos\left(\frac{\mathbf{e}_{g,k1} \cdot \mathbf{e}_{g,k2}}{\|\mathbf{e}_{g,k1}\| \cdot \|\mathbf{e}_{g,k2}\|}\right), \quad (6)$$

where $\cdot$ denotes dot product, computing the angle between edges and providing local curvature information.

These features are concatenated to form the 2D simplicial feature vector:

$$\mathbf{S}_2 = [\mathbf{normal}, \mathbf{A}, \theta] \in \mathbb{R}^{G \times K \times 5}, \quad (7)$$

encoding surface characteristics of the local region.

Finally, we integrate all three simplicial features to construct the complete topological primitive representation:

$$\mathbf{f}_{g,k} = [\mathbf{S}_0, \mathbf{S}_1, \mathbf{S}_2] \in \mathbb{R}^{G \times K \times 14}, \quad (8)$$

explicitly modeling multi-scale topological structure from points to edges to surfaces.

### 3.3.2. GEOMETRY CONTROLLER

The Geometry Controller intelligently fuses topological features with semantic features, dynamically adjusting their contribution to adapt to diverse point cloud structures and task requirements. This adaptive mechanism balances local geometric information with global semantic features, enhancing versatility across scenarios.

We employ a projection function to inject topological primitives into the semantic feature space. First, a $1 \times 1$ convolution maps primitives to latent space, generating topological weights:

$$W_{g,k}^{top} = \text{Conv}_{1\times1}(\mathbf{f}_{g,k}) \in \mathbb{R}^{G \times K \times d}, \quad (9)$$

where $d$ represents feature dimension. The topological aggregation function is:

$$\mathcal{F}_{g,top} = \mathcal{A}(\{W_{g,k}^{top} \odot \mathbf{n}_{g,k} | \mathbf{n}_{g,k} \in \mathcal{N}(\mathbf{c}_g)\}) \in \mathbb{R}^{G \times d}, \quad (10)$$

where $\mathcal{A}$ denotes the aggregation function (max pooling for permutation invariance), $\odot$ represents element-wise product, and $\mathcal{N}(\mathbf{c}_g)$ defines the local neighborhood.

We introduce a learnable geometric control parameter $\alpha$ to regulate the fusion ratio:

$$\mathcal{F}_{g,global} = \alpha \odot \mathcal{F}_{g,pointnet} + (1 - \alpha) \odot \mathcal{F}_{g,top}, \quad (11)$$

where $\mathcal{F}_{g,pointnet} \in \mathbb{R}^{G \times d}$ represents semantic features from the pre-trained backbone. The resulting $\mathcal{F}_{g,global}$ constitutes the ToInjection output, subsequently passed to downstream blocks. This dynamic fusion mechanism enables automatic balancing between geometric and semantic information based on input characteristics, significantly enhancing adaptability across point cloud understanding tasks.

### 3.4. Topology Transfer Module

The Topology Transfer module (ToTransfer) constitutes the second core component, designed to propagate topological primitive features $\mathbf{f}_{g,k}$ generated by ToInjection across multiple layers of the pre-trained model. This module establishes pathways for geometric information flow between Transformer encoder layers, ensuring topological information utilization throughout feature extraction. The design addresses the limitation that existing methods often confine geometric information to input or shallow layers, failing to maintain this critical information in deeper layers.

### 3.4.1. MAPPING MECHANISM OF TOPOLOGICAL PRIMITIVES

The local topological primitive features $\mathbf{f}_{g,k}$ encapsulate rich geometric information; however, their raw form is incompatible with direct Transformer integration. Therefore, we design a specialized mapping mechanism to transform these primitives into a compatible representation format. To enhance discriminative power, we implement a processing pipeline with max pooling and Softmax normalization:

$$G_{top} = \text{MaxPool}(\text{MLP}_{group}(\mathbf{f}_{g,k})) \in \mathbb{R}^{G \times d}, \quad (12)$$

$$G_{top}^{norm} = \text{Softmax}(G_{top}) \in \mathbb{R}^{G \times d}. \quad (13)$$

This normalization strategically weights each feature dimension while preserving the relative importance of underlying topological structure, creating a representation that effectively interacts with the Transformer's attention mechanism.

### 3.4.2. CROSS-LAYER TOPOLOGICAL PROMPT TRANSFER

ToTransfer injects normalized topological prompt features $G_{top}^{norm}$ into each Transformer encoder layer through a systematic cross-layer transfer mechanism. Let the input to the $l$-th layer be $X_l \in \mathbb{R}^{G \times d}$, with output $X_{l+1} \in \mathbb{R}^{G \times d}$. The topological prompt transfer process is:

$$X_l' = X_l \odot (1 + G_{top}^{norm}) \in \mathbb{R}^{G \times d}, \quad (14)$$

*Table 1.* Classification on three variants of the ScanObjectNN(Uy et al., 2019) and the ModelNet40(Wu et al., 2015), including the number of trainable parameters (Param) and overall accuracy (Acc). We report ScanObjectNN(Uy et al., 2019) and ModelNet40(Wu et al., 2015) results without voting.

| Method | Reference | Param.(M) ↓ | ScanObjectNN | | | ModelNet40 |
|--------|-----------|-------------|--------------|--------------|--------------|-------------|
| | | | OBJ_BG | OBJ_ONLY | PB_T50_RS | Acc. (%) ↑ |
| *Self-Supervised Representation Learning (Full Fine-Tuning)* | | | | | | |
| OcCo(Wang et al., 2021) | ICCV'21 | 22.1 | 84.85 | 85.54 | 78.79 | 92.1 |
| Point-BERT(Yu et al., 2022) | CVPR'22 | 22.1 | 87.43 | 88.12 | 83.07 | 92.7 |
| MaskPoint(Liu et al., 2022) | ECCV'22 | 22.1 | 89.70 | 89.30 | 84.60 | 93.8 |
| Point-MAE(Pang et al., 2022) | ECCV'22 | 22.1 | 90.02 | 88.29 | 85.18 | 93.2 |
| Point-M2AE(Zhang et al., 2022) | NeurIPS'22 | 15.3 | 91.22 | 88.81 | 86.43 | 93.4 |
| ReCon(Qi et al., 2023) | ICML'23 | 43.6 | 94.15 | 93.12 | 89.73 | 93.9 |
| PointGPT-L(Chen et al., 2023) | NeurIPS'23 | 360.5 | 97.20 | 96.60 | 93.40 | 94.1 |
| *Self-Supervised Representation Learning (Parameter-Efficient Fine-Tuning)* | | | | | | |
| Point-MAE(Pang et al., 2022) | ECCV'22 | 22.1(100%) | 90.02 | 88.29 | 85.18 | 93.2 |
| +IDPT(Zha et al., 2023) | ICCV'23 | 1.7(7.69%) | 91.22(+1.20) | 90.02(+1.73) | 84.94(-0.24) | 93.3(+0.1) |
| +DAPT(Zhou et al., 2024) | CVPR'24 | 1.1(4.97%) | 90.88(+0.86) | 90.19(+1.90) | 85.08(-0.10) | 93.5(+0.3) |
| +Point-PEFT(Tang et al., 2024) | AAAI'24 | 0.7(3.17%) | 89.33(-0.69) | 88.98(+0.69) | 84.42(-0.76) | 94.2(+1.0) |
| +PointGST(Liang et al., 2025) | TPAMI'25 | 0.6(2.77%) | 91.74(+1.72) | 90.19(+1.90) | 85.29(+0.11) | 93.5(+0.3) |
| +PointLoRA(Wang et al., 2025b) | CVPR'25 | 0.77(3.43%) | 90.71(+0.69) | 89.33(+1.04) | 85.53(+0.35) | 93.3(+0.1) |
| +GAPrompt(Ai et al., 2025) | ICML'25 | 0.6(2.71%) | 91.91(+1.89) | 90.19(+1.90) | 85.57(+0.39) | 94.2(+1.0) |
| +**TopAdapter** | **Ours** | **0.5(2.26%)** | **92.08(+2.06)** | **91.74(+3.45)** | **85.58(+0.40)** | **94.2(+1.0)** |
| ReCon(Qi et al., 2023) | ICML'23 | 43.6(100%) | 94.15 | 93.12 | 89.73 | 93.9 |
| +IDPT(Zha et al., 2023) | ICCV'23 | 1.7(3.90%) | 93.29(-0.86) | 91.57(-1.55) | 87.27(-2.46) | 93.4(-0.5) |
| +DAPT(Zhou et al., 2024) | CVPR'24 | 1.1(2.52%) | 94.32(+0.17) | 92.43(-0.69) | 89.38(-0.35) | 93.5(-0.4) |
| +Point-PEFT(Tang et al., 2024) | AAAI'24 | 0.7(1.61%) | 92.94(-1.21) | 91.57(-1.55) | 89.07(-0.66) | 93.8(-0.1) |
| +PointGST(Liang et al., 2025) | TPAMI'25 | 0.6(1.37%) | 94.49(+0.34) | 92.94(-0.18) | 89.49(-0.24) | 93.6(-0.3) |
| +GAPrompt(Ai et al., 2025) | ICML'25 | 0.6(1.37%) | 94.49(+0.34) | 92.60(-0.52) | 89.76(+0.03) | 94.0(+0.1) |
| +**TopAdapter** | **Ours** | **0.5(1.15%)** | **94.53(+0.37)** | **93.46(+0.34)** | **89.82(+0.09)** | **94.0(+0.1)** |
| PointGPT-L(Chen et al., 2023) | NeurIPS'23 | 360.5(100%) | 97.20 | 96.60 | 93.40 | 94.1 |
| +IDPT(Zha et al., 2023) | ICCV'23 | 10.0(2.77%) | 98.11(+0.91) | 96.04(-0.56) | 92.99(-0.41) | 93.4(-0.7) |
| +DAPT(Zhou et al., 2024) | CVPR'24 | 4.2(1.17%) | 98.11(+0.91) | 96.21(-0.39) | 93.02(-0.38) | 94.2(+0.1) |
| +Point-PEFT(Tang et al., 2024) | AAAI'24 | 3.1(0.86%) | 97.76(+0.56) | 96.21(-0.39) | 93.11(-0.29) | 93.9(-0.2) |
| +PointGST(Liang et al., 2025) | TPAMI'25 | 2.4(0.67%) | 98.97(+1.77) | 97.59(+0.99) | 94.83(+1.43) | 94.8(+0.7) |
| +GAPrompt(Ai et al., 2025) | ICML'25 | 2.0(0.55%) | 98.97(+1.77) | 96.73(+0.13) | 94.41(+0.91) | **96.2(+2.1)** |
| +**TopAdapter** | **Ours** | **1.9(0.53%)** | **99.48(+2.28)** | **97.76(+1.16)** | **94.97(+1.57)** | 94.5(+0.4) |

modulating original features with topological information through element-wise multiplication with a shifted normalized prompt.

To further enhance geometric perception, ToTransfer introduces a lightweight adapter module for each Transformer block. The updated processing sequence becomes:

$$X'_{l+1} = X'_l + \text{Drop}(\text{Attn}(\text{Norm1}(X'_l))) \in \mathbb{R}^{G \times d}, \quad (15)$$

$$X''_{l+1} = X'_{l+1} + \text{Drop}\big(\text{MLP}(\text{Norm2}(X'_{l+1}))\big) + H_{\text{adapter}} \in \mathbb{R}^{G \times d}, \quad (16)$$

$$H_{adapter} = \text{MLP}_{adapter}(X'_l) \in \mathbb{R}^{G \times d}, \quad (17)$$

where $\text{Drop}(\cdot)$ represents Dropout for regularization; $\text{Attn}(\cdot)$ denotes multi-head self-attention; $\text{Norm1}(\cdot)$ and $\text{Norm2}(\cdot)$

are layer normalization; $\text{MLP}(\cdot)$ represents the standard feedforward network; and $\text{MLP}_{adapter}(\cdot)$ consists of two fully connected layers with GELU activation, introducing a parameter-efficient adaptation path.

## 4. Experiments

### 4.1. Datasets and Evaluation Metrics

We conduct comprehensive experiments to evaluate the efficacy of **TopAdapter** across multiple benchmarks. Our evaluation utilizes three widely-recognized datasets:

**ScanObjectNN** (Uy et al., 2019) represents a challenging real-world 3D dataset comprising approximately 15,000 indoor point cloud objects across 15 categories. These objects, acquired through direct scanning, exhibit authen-

tic real-world characteristics including background clutter and partial occlusion. We conduct experiments on three variants of increasing complexity: OBJ_BG (objects with background), OBJ_ONLY (objects without background), and PB_T50_RS (objects with background, occlusion, and deformation). Each point cloud sample consists of 2048 points.

**ModelNet40** (Wu et al., 2015) constitutes a synthetic 3D CAD model dataset encompassing 12,311 complete models across 40 categories. These point clouds are characterized by completeness, uniformity, and minimal noise, resulting in a relatively straightforward classification task.

**ShapeNetPart** (Yi et al., 2016) serves as a widely adopted benchmark for part segmentation, containing 16,881 synthetic objects across 16 object categories annotated with 50 part categories. We utilize this dataset to assess our model's capacity to capture fine-grained structural details and perform accurate part segmentation.

**Evaluation Metrics**: For classification tasks, we employ overall accuracy (OA) as the primary metric, representing the ratio of correctly classified instances to total instances. For part segmentation, we utilize mean Intersection over Union (mIoU) to quantify the overlap between predicted and ground truth labels, calculating both class-average mIoU and instance-average mIoU for comprehensive performance assessment.

### 4.2. Implementation Details

For fair comparison, we adopt identical data augmentation strategies as the original full fine-tuning methods. We employ the AdamW optimizer with a learning rate of 5e-4 and weight decay of 0.05. Classification models are trained for 300 epochs with cosine learning rate scheduling and a 10-epoch warm-up, while segmentation tasks use a reduced learning rate of 2e-4. Our **TopAdapter** incorporates 0D, 1D, and 2D simplices to capture topology information of point cloud. The geometric controller parameter $\alpha$ is set to 0.6, and the number of neighbors $K$ is set to 30. We use a batch size of 32 for all experiments and apply standard data augmentation techniques including random rotation and scaling. All experiments are conducted on an NVIDIA RTX 3090 GPU using PyTorch 1.13.1.

### 4.3. Experimental Results and Analysis

#### 4.3.1. RESULTS ON SCANOBJECTNN

On the challenging real-world ScanObjectNN (Uy et al., 2019) dataset, **TopAdapter** demonstrates superior performance with minimal parameter overhead. As shown in Table 1, when applied to Point-MAE (Pang et al., 2022), **TopAdapter** achieves significant improvements with only

*Table 2.* Performance comparison on the ShapeNetPart (Yi et al., 2016) for part segmentation.

| Methods | Publication | TP | Cls. mIoU (%) | Inst. mIoU (%) |
|---|---|---|---|---|
| Traditional Supervised Learning Only | | | | |
| PointNet (Qi et al., 2017a) | CVPR'17 | - | 80.39 | 83.7 |
| PointNet++ (Qi et al., 2017b) | NeurIPS'17 | - | 81.85 | 85.1 |
| DGCNN (Wang et al., 2019) | TOG'19 | - | 82.33 | 85.2 |
| APES (Wu et al., 2023) | CVPR'23 | - | 83.67 | 85.8 |
| Self-Supervised Representation Learning (Full Fine-Tuning) | | | | |
| OcCo (Wang et al., 2021) | ICCV'21 | 27.09 M | 83.42 | 85.1 |
| MaskPoint (Liu et al., 2022) | ECCV'22 | - | 84.60 | 86.0 |
| Point-BERT (Yu et al., 2022) | CVPR'22 | 27.09 M | 84.11 | 85.6 |
| Point-MAE (Pang et al., 2022) | ECCV'22 | 27.06 M | 84.19 | 86.1 |
| ACT (Dong et al., 2022) | ICLR'23 | 27.06 M | 84.66 | 86.1 |
| Self-Supervised Representation Learning (Parameter-Efficient Fine-Tuning) | | | | |
| RECON (Qi et al., 2023) (Full-FT) | ICML'23 | 27.06 M | 84.52 | 86.1 |
| +IDPT (Zha et al., 2023) | ICCV'23 | 5.69 M | 83.66 | 85.7 |
| +DAPT (Zhou et al., 2024) | CVPR'24 | 5.65 M | 83.87 | 85.7 |
| +PointGST(Liang et al., 2025) | TPAMI'25 | 5.59 M | 83.98 | 85.8 |
| +PointLoRA(Wang et al., 2025b) | CVPR'25 | 5.63 M | 83.98 | 85.4 |
| +**TopAdapter** | **Ours** | **5.58 M** | **83.99** | **85.8** |

0.5M parameters (2.26%): increasing accuracy by 2.06% on OBJ_BG (92.08%), 3.45% on OBJ_ONLY (91.74%), and 0.40% on the challenging PB_T50_RS variant (85.58%), outperforming all competing PEFT methods. With Re-Con (Qi et al., 2023), **TopAdapter** maintains this advantage, achieving 94.53% and 93.46% on OBJ_BG and OBJ_ONLY respectively. Most impressively, when applied to PointGPT-L (Chen et al., 2023), **TopAdapter** requires merely 1.9M parameters (0.53%) yet delivers state-of-the-art performance across all variants: 99.48% on OBJ_BG (+2.28%), 97.76% on OBJ_ONLY (+1.16%), and 94.97% on PB_T50_RS (+1.57%), even exceeding full fine-tuning results. These improvements highlight the effectiveness of our topology-aware design in capturing essential geometric structures.

#### 4.3.2. RESULTS ON MODELNET40

On the ModelNet40 (Wu et al., 2015) dataset, **TopAdapter** maintains competitive performance. With Point-MAE, it achieves 94.2% accuracy (+1.0% over baseline), and with ReCon, it reaches 94.0%, surpassing the original 93.9%. For PointGPT-L, **TopAdapter** achieves 94.5%, exceeding the full fine-tuning baseline of 94.1%. While PointGST achieves marginally higher accuracy (94.8%) in this specific case, **TopAdapter** consistently delivers strong performance with fewer parameters, demonstrating excellent adaptability across different model architectures and confirming its value for resource-constrained applications.

#### 4.3.3. RESULTS ON SHAPENETPART

**TopAdapter** demonstrates competitive performance on the ShapeNetPart (Yi et al., 2016) segmentation benchmark, as illustrated in Table 2. When applied to Re-Con, **TopAdapter** achieves 83.99% class-average mIoU and 85.8% instance-average mIoU, establishing itself as

*Table 3.* Comparison with general PEFT methods and point cloud-specific approaches on ScanObjectNN(Uy et al., 2019).

| Methods | Publication | TP | PB-T50-RS |
|---|---|---|---|
| Point-MAE (Pang et al., 2022) | ECCV'22 | 22.1 M | 85.18 |
| Linear probing | - | 0.3 M | 75.99 |
| + Adapter (Houlsby et al., 2019) | ICML'19 | 0.9 M | 83.93 |
| + Prefix tuning (Li & Liang, 2021) | ACL'21 | 0.7 M | 77.72 |
| + BitFit (Zaken et al., 2022) | ACL'21 | 0.3 M | 82.62 |
| + LoRA (Hu et al., 2022) | ICLR'22 | 0.9 M | 81.74 |
| + VPT-Deep (Jia et al., 2022) | ECCV'22 | 0.4 M | 81.09 |
| + AdapterFormer (Chen et al., 2022) | NeurIPS'22 | 0.9 M | 83.45 |
| + SSF (Lian et al., 2022) | NeurIPS'22 | 0.4 M | 82.58 |
| + IDPT (Zha et al., 2023) | ICCV'23 | 1.7 M | 84.94 |
| + DAPT (Zhou et al., 2024) | CVPR'24 | 1.1 M | 85.08 |
| + PointGST(Liang et al., 2025) | TPAMI'25 | 0.6 M | 85.29 |
| + PointLoRA(Wang et al., 2025b) | CVPR'25 | 0.77 M | 85.53 |
| + **TopAdapter** | **Ours** | **0.5 M** | **85.58** |

*Table 4.* Ablation study on **TopAdapter**'s key components.

| Model Configuration | Params (M) | Accuracy (%) |
|---|---|---|
| PointGPT-L (Full-FT) | 360.5 | 93.40 |
| + ToInjection Only | 1.5 | 94.26 |
| + ToTransfer Only | 1.3 | 94.13 |
| + **TopAdapter** | 1.9 | 94.97 |

*Table 5.* Ablation study on different simplex dimension combinations.

| Simplex Combination | Params (M) | Accuracy (%) |
|---|---|---|
| Baseline (No Simplex) | 1.1 | 93.62 |
| 0D Simplex Only | 1.4 | 94.15 |
| 1D Simplex Only | 1.4 | 94.08 |
| 2D Simplex Only | 1.5 | 94.21 |
| 0D + 1D | 1.7 | 94.53 |
| 0D + 2D | 1.7 | 94.61 |
| 1D + 2D | 1.7 | 94.49 |
| 0D + 1D + 2D | 1.9 | 94.97 |

the top-performing parameter-efficient method for this task. While these results trail the full fine-tuning baseline (84.52%/86.1%) by a small margin, they represent clear improvements over IDPT (83.66%/85.7%) and DAPT (83.87%/85.7%). Notably, **TopAdapter** achieves results comparable to or better than PointGST (83.98%/85.8%) and PointLoRA (83.98%/85.4%) while requiring fewer parameters (5.58M), highlighting superior parameter efficiency. These results confirm **TopAdapter**'s effectiveness at capturing fine-grained local geometric features essential for part segmentation, further validating the advantage of our topology-aware approach for detailed 3D tasks.

#### 4.3.4. COMPARISON WITH GENERAL PEFT METHODS

We benchmark **TopAdapter** against PEFT methods from NLP and 2D vision on the challenging PB_T50_RS variant of ScanObjectNN (Uy et al., 2019) using Point-MAE (Pang et al., 2022) as the baseline (Table 3). General-purpose PEFT methods such as Adapter (83.93%), AdapterFormer (83.45%), and LoRA (81.74%) demonstrate limited effectiveness when applied to point cloud data, despite their success in other domains. This performance gap highlights the unique challenges of 3D geometric data. In contrast, point cloud-specific methods including IDPT (84.94%), DAPT (85.08%), and PointGST (85.29%) offer better performance. **TopAdapter** achieves the highest accuracy (85.58%) while requiring the fewest parameters (0.5M), outperforming both general parameter-efficient fine-tuning approaches and specialized point cloud methods. This superior performance stems from **TopAdapter**'s topology-based design that effectively captures multi-scale geometric structures through 0D, 1D, and 2D simplices.

### 4.4. Ablation Studies

#### 4.4.1. IMPACT OF DIFFERENT MODULES

We conduct ablation studies on PointGPT-L using the challenging PB_T50_RS variant of ScanObjectNN. Table 4 demonstrates the contribution of each component in **TopAdapter**. Both individual modules outperform full fine-tuning (93.40%) while using less than 0.5% of the parameters: ToInjection achieves 94.26% by effectively extracting topological features, while ToTransfer reaches 94.13% through cross-layer geometric information propagation. The complete **TopAdapter** combines these complementary strengths, achieving 94.97% accuracy with only 1.9M parameters, confirming the effectiveness of our design for capturing and utilizing point cloud topology.

#### 4.4.2. IMPACT OF SIMPLEX DIMENSIONS

Table 5 demonstrates how different simplex dimensions contribute to **TopAdapter**'s performance. Individual simplex representations each improve over the baseline (93.62%), with 2D simplices (94.21%) slightly outperforming 0D (94.15%) and 1D (94.08%) variants, confirming the value of surface-level geometric information. Dual-dimension combinations further enhance performance, with 0D+2D achieving 94.61%, suggesting the complementary nature of positional and surface information. The complete model incorporating all three dimensions (0D+1D+2D) achieves the highest accuracy (94.97%), validating our theoretical foundation that multi-scale topological primitives comprehensively capture point cloud geometry from positions to connectivity and surface properties.

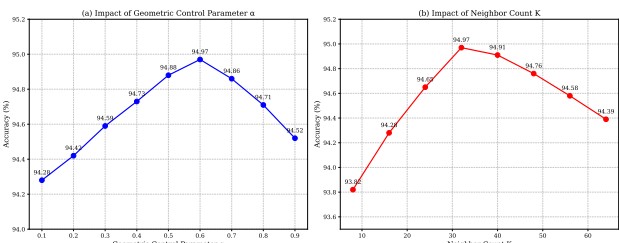

*Figure 4.* Performance sensitivity analysis of **TopAdapter** to (a) geometric control parameter $\alpha$ and (b) neighbor count $K$.

### 4.4.3. SENSITIVITY ANALYSIS OF GEOMETRIC CONTROL PARAMETER $\alpha$

Fig. 4(a) illustrates the impact of the geometric control parameter $\alpha$, which balances topological and semantic features. Performance gradually improves as $\alpha$ increases from 0.1 (94.28%) to the optimal value of 0.6 (94.97%), then declines with further increases. This trend confirms that while topological features provide valuable geometric information, semantic features remain essential, with the optimal balance achieved at $\alpha = 0.6$. This finding underscores the importance of our adaptive fusion mechanism in **TopAdapter**.

### 4.4.4. IMPACT OF NEIGHBORHOOD SIZE $K$

Fig. 4(b) demonstrates how the number of neighbors $K$ affects model performance. With insufficient neighbors ($K = 10$), performance is limited (93.82%) as the model captures only sparse local structures. Performance improves significantly as $K$ increases to 30 (94.97%), then gradually decreases with larger neighborhoods. This pattern indicates that an appropriate neighborhood size is crucial for constructing effective simplicial complexes—too few neighbors provide insufficient geometric context, while too many introduce noise from distant, less relevant points.

## 5. Conclusion

In this paper, we introduced **TopAdapter**, a novel parameter-efficient fine-tuning framework that leverages topological principles to enhance point cloud understanding. By incorporating simplices as fundamental building blocks, our approach effectively captures multi-scale geometric structures—from positions (0D) to connectivity (1D) and surface properties (2D)—addressing a critical limitation in existing PEFT methods. Through Topology Injection and Topology Transfer modules, **TopAdapter** achieves remarkable parameter efficiency while maintaining or surpassing full fine-tuning performance. Extensive experiments demonstrate that with only 0.5M parameters, **TopAdapter** significantly outperforms existing methods across ScanObjectNN, ModelNet40, and ShapeNetPart, establishing a new paradigm for efficient 3D point cloud processing.

## Impact Statement

This paper presents work whose goal is to advance the field of machine learning in 3D point cloud understanding through parameter-efficient fine-tuning. Our **TopAdapter** framework has potential societal benefits including reduced computational costs and energy consumption for training 3D models, contributing to more sustainable AI practices, and democratizing access to advanced 3D vision technology for researchers with limited resources. However, as with any advancement in 3D computer vision, there exist potential concerns regarding privacy and surveillance applications. We encourage responsible deployment of such technologies with appropriate consideration of ethical guidelines and user consent. Overall, we believe this work contributes positively to the field while the potential risks are well-established concerns common to 3D vision research broadly.

## Acknowledgments

This work was supported in part by the European Union's Horizon 2024 Research and Innovation Programme for the Marie Skłodowska-Curie Actions under Grant No. 101211118. Shuting He was sponsored by Shanghai Pujiang Programme 24PJD030 and Natural Science Foundation of Shanghai 25ZR1402138.

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
