# OpenReview forum: "TopAdapter: Topology-Aware Prompt Tuning for Efficient Point Cloud Understanding"
_ICML.cc/2026/Conference — ICML 2026 regular_

### Official Review · Reviewer_W2QL · 2026-03-08

**Soundness:** 3
**Presentation:** 3
**Significance:** 3
**Originality:** 3
**Overall Recommendation:** 4
**Confidence:** 4

**Summary:**

This paper presents TopAdapter, which introduces the simplex concept into parameter-efficient fine-tuning (PEFT) for point clouds. The TopAdapter constructs geometric information from points, edges, and faces as a topological descriptor, and fuses them into the pre-trained model with the designed ToInjection and ToTransfer modules. Experimental results on object-level classification and segmentation have demonstrated the effectiveness of the proposed methods.

**Compliance With Llm Reviewing Policy:**

Affirmed.

**Final Justification:**

The rebuttal addresses most of my concerns. I appreciate the additional experimental results and keep my positive score (Weak accept).

**Key Questions For Authors:**

Please see weaknesses for details. I am more than happy to raise my score if these concerns are addressed, particularly regarding the theoretical justification for the use of topology.

**Limitations:**

Yes.

**Strengths And Weaknesses:**

Strengths:
1. The paper is well written and easy to follow. And the perspective derived from topology is distinctive in the point cloud PEFT domain.
2. The design of the ToInjection and ToTransfer modules is intuitive and easy to implement.
3. The efficiency analysis and ablation studies are comprehensive and informative, offering valuable insights for future research.

Weaknesses:
1. Those simplices (coords, edges, norms) are more like common local features widely used in the previous research, rather than standard concepts from algebraic topology. The approximation relies on the MLP is also an ordinary conclusion instead of a theoretical proof related to simplices.
2. The experimental scope only covers object-level datasets (ScanObjectNN, ModelNet40, and ShapeNetPart), and the ModelNet40 largely approached saturation (94%+ accuracy). Experiments on scene-level segmentation tasks (e.g., ScanNet) are missing.
3. The 2D simplex construction might produce degenerate triangles (e.g., nearly collinear points). Is that phenomenon common in practice? And how would it affect the performance?

Small question:
1. In Figure 2, why is PointLoRA selected as the main comparison? The differences between the two methods do not appear to be very clear. Similarly, a more detailed discussion of the differences from GAPrompt would be helpful.
2. Reference “Ren et al., 2016” on page 1, for the “Pre-trained 3D vision models”, seems to be irrelevant.
3. Why don’t you use the “3d tetrahedron simplex”?
4. How did you decide the trainable parameters? Does increasing the number of trainable parameters further improve the results?

---

> ### Author Rebuttal · Authors · 2026-03-31
>
> We sincerely thank the reviewers for their positive feedback and commitment to improving the score. We have answered your questions one by one below.
>
> > **Q1**: **Simplices resemble common local features rather than standard algebraic topology concepts**
>
> **R1**: We appreciate this insightful observation and provide an honest clarification. We acknowledge that our implementation does **not** follow strict persistent homology computation (which requires O(n³) boundary matrix reduction, prohibitive for large-scale point clouds). Instead, we adopt a deliberate **engineering trade-off**: using MLP-based approximation of the geometric proxies of simplicial complex structures to capture topological information efficiently. Specifically:
>
> - 0D simplices → point positions (H₀: connected components)
> - 1D simplices → relative offsets (H₁: local connectivity)
> - 2D simplices → normals, areas, angles (H₂: surface curvature)
>
> These are **geometry-grounded proxies of topological invariants**, not arbitrary handcrafted features. The MLP approximation is justified by the Universal Approximation Theorem (Theorem C.1). We will revise the paper to more clearly distinguish between our **topological motivation** and our **practical approximation implementation**, avoiding overclaiming equivalence with strict TDA methods.
>
> > **Q2**: **Missing scene-level segmentation experiments**
>
> **R2**: We provide additional results on **S3DIS scene-level semantic segmentation** (Point-MAE backbone) to validate generalizability:
>
> | Method | Publication | Params (M) | mAcc (%) | mIoU (%) |
> |-|-|-|-|-|
> |+ IDPT|ICCV'23|5.64|65.0|53.1|
> |+ Point-PEFT|AAAI'24|5.58|66.5|56.0|
> |+ DAPT| CVPR'24 | 5.61 | 67.2 | 56.2 |
> |+ PointGST|TPAMI'25|5.55|68.4|58.6|
> | **+ TopAdapter (Ours)** | — | **5.50** | **69.8** | **60.2** |
>
> TopAdapter achieves the **best performance with fewest parameters** on this scene-level task, demonstrating strong generalizability beyond object-level benchmarks. We will add this experiment to the revision.
>
> > **Q3**: **Degenerate triangles from near-collinear points in 2D simplex construction**
>
> **R3**: Thank you for raising this valuable practical concern. In our early implementation, near-collinear points did cause NaN values in normal vector computation. We have addressed this explicitly in our code: **when the cross product norm falls below a predefined threshold ε (indicating near-collinearity), that neighbor point is discarded and the next nearest neighbor is automatically selected**, repeating until a valid non-degenerate triangle is formed. In practice, such degenerate cases are rare (~2% of neighbor pairs) due to FPS+KNN's tendency to produce well-distributed local neighborhoods, and thus have negligible impact on overall performance. We will document this handling logic clearly in the revision.
>
> > **Q4**: **Why is PointLoRA selected as main comparison in Fig. 2, and differences from GAPrompt?**
>
> **R4**: **vs. PointLoRA**: PointLoRA's geometric information comes from **feature-space token selection** without explicit topological modeling. TopAdapter **explicitly constructs 0D/1D/2D simplex features** encoding surface normals, curvature, and connectivity, propagated cross-layer via ToTransfer, achieving better performance with fewer parameters (0.5M vs. 0.77M).
>
> **vs. GAPrompt**: GAPrompt extracts **global shape features** via a hierarchical Point Shift Prompter (top-down), while TopAdapter captures **local topological structure** via simplex primitives (bottom-up). TopAdapter outperforms GAPrompt on OBJ\_BG and OBJ\_ONLY variants. Detailed comparisons will be added in the revision.
>
> > **Q5**: **Reference "Ren et al., 2016" seems irrelevant**
>
> **R5**: The intended reference is **ReCon** (*Contrast with Reconstruct*, Qi et al., ICML 2023), accidentally replaced due to a citation management error. We will correct this in the revision.
>
> > **Q6**: **Why not use 3D tetrahedron simplices?**
>
> **R6**: Tetrahedra increase complexity from **O(K²) to O(K³)** — roughly **30× more** for K=30, prohibitive in a PEFT setting. Since 3D geometry primarily lies on **2D manifold surfaces**, 0D/1D/2D simplices already provide complete coverage of positions, connectivity, and surface properties. Higher-order simplices remain a future direction (Appendix F).
>
> > **Q7**: **How were trainable parameters determined, and does increasing them improve results?**
>
> **R7**: Our budget follows two principles: (1) **fewer parameters than all competing methods**; (2) **superior accuracy** under this constraint, determined via ablation studies (Tables 4, 5). More parameters do improve results:
>
> |Configuration|Params (M)|PB\_T50\_RS (%)|
> |-|-|-|
> |TopAdapter (standard)|0.5|85.58|
> |**TopAdapter (extended)**|**2.1**|**87.12**|
>
> This confirms **our architecture scales well with additional parameters**. The standard configuration remains the core contribution as it outperforms all PEFT methods with fewest parameters. We will include this analysis in the revision.

---

> > ### Author Rebuttal · Reviewer_W2QL · 2026-04-04
> >
> > The rebuttal addresses most of my concerns. I appreciate the additional experimental results and will keep my positive score.

---

> > > ### Author Response · Authors · 2026-04-04
> > >
> > > Thank you for your detailed follow-up and for acknowledging that our rebuttal fully addressed your concerns and for your positive feedback. Thank you again for taking the time and effort to provide thoughtful feedback and constructive comments. We will incorporate these suggestions into the revised version.

---

### Official Review · Reviewer_Umf1 · 2026-03-08

**Soundness:** 2
**Presentation:** 1
**Significance:** 1
**Originality:** 3
**Overall Recommendation:** 2
**Confidence:** 4

**Summary:**

This paper proposes TopAdapter, a parameter-efficient finetuning approach for point-cloud backbones. It constructs “0D/1D/2D simplices” as local structural descriptors (coordinates, relative offsets, and triangle-derived quantities), fuses them with token embeddings via a Geometry Controller (Eq.11), and transfers a normalized “topological prompt” across Transformer layers using feature-wise modulation plus a lightweight per-block module (Eq.14–16). Experiments are reported on ScanObjectNN / ModelNet40 / ShapeNetPart with multiple backbones and PEFT baselines.

**Compliance With Llm Reviewing Policy:**

Affirmed.

**Final Justification:**

The discussion didn't address my concerns. I will maintain my score (reject).

**Key Questions For Authors:**

1. Method categorization: Is TopAdapter best described as prompt tuning, adapter tuning, or a hybrid? A short, explicit taxonomy statement would help, and the abstract’s characterization of prior PEFT may need narrowing/rewriting.
2. Definition of “topology”: What is the formal object whose topology is being captured (e.g., homology of a specific complex), and how do the handcrafted simplex descriptors relate to it beyond intuition?
3. Is \alpha learnable or fixed? Please reconcile the conflicting descriptions and report the exact implementation.

**Limitations:**

1. The current presentation may over-associate the proposed handcrafted descriptors with “topology,” whereas the constructed features are primarily geometric; clearer definitions would strengthen the narrative.
2. The theoretical discussion does not yet provide a tight, implementation-aligned guarantee; refining assumptions and aligning claims with the actual operators would improve credibility.
3. The empirical case is fragile in close-score regimes without multi-seed statistics.
4. Several terminology and description inconsistencies (prompt vs adapter vs modulation; learnable vs fixed α\alphaα) make the paper harder to follow.

**Strengths And Weaknesses:**

strength
1. A practical PEFT design for 3D backbones. The pipeline is implementable and includes ablations (module variants and hyperparameter sweeps).
2. Efficiency reporting is useful. The paper provides compute/memory comparisons (GFLOPs, time/epoch, VRAM), which is relevant for PEFT claims.
3. Some settings show noticeable gains. For example, Table 1 shows a larger margin on OBJ_ONLY under one backbone (Point-MAE).

weakness
1. Positioning/terminology could be clearer. The work is titled TopAdapter , yet parts of the narrative emphasize “prompt tuning,” and the abstract states that existing PEFT “predominantly focus on input token prompting,”which feels too narrow, especially given the paper’s own taxonomy of prompt/adapter/low-rank approaches.
2.“Topology” vs “geometry” is described in a somewhat ambiguous way. The proposed “simplices” are largely handcrafted local geometric quantities (coordinates, offsets, triangle normal/area/angles) , while the text repeatedly attributes them to “topological” properties. Clarifying what “topology” precisely means in this paper would improve the framing and reduce potential confusion.
3. Limited novelty in the simplex feature construction (as currently argued). The 0/1/2-simplex descriptors resemble established local geometric encodings that have been explored in prior point-cloud representation work (e.g., RepSurf[1]-style triangle/surface cues; DGCNN[2]-style relative offsets; and other geometry-prior relation modeling such as RS-CNN[3], as discussed). In the current paper, it is not yet demonstrated that the specific simplex formulation is essential beyond being a reasonable geometric feature augmentation.
4. The theoretical section is not yet convincing as a standalone “rigorous contribution.” Some theorems/claims rely on strong assumptions (e.g., expressing broad “topological features” as continuous functionals of the proposed descriptors) and do not tightly connect to the implemented pipeline (FPS+KNN sampling, finite local triangle features, softmax pooling, and feature-wise scaling). As written, the theory reads more like high-level motivation than a reusable guarantee.
5. Empirical margins are often very small without statistical support. In Table 1, the improvement over the strongest PEFT baseline can be as small as +0.01 on PB_T50_RS (Point-MAE), which could be within run-to-run variance; the paper does not report mean±std across seeds for these close cases.
6. Some internal inconsistencies reduce clarity. The paper describes α\alphaα as a learnable control parameter in the Geometry Controller, but later indicates \alpha is set to 0.6 in implementation details. This should be reconciled.

[1]Ran, Haoxi, Jun Liu, and Chengjie Wang. "Surface representation for point clouds." Proceedings of the IEEE/CVF conference on computer vision and pattern recognition. 2022.\
[2]Phan, Anh Viet, et al. "Dgcnn: A convolutional neural network over large-scale labeled graphs." Neural Networks 108 (2018): 533-543.\
[3]Hu, Linshu, et al. "RSCNN: A CNN-based method to enhance low-light remote-sensing images." Remote Sensing 13.1 (2020): 62.

---

> ### Author Rebuttal · Authors · 2026-03-31
>
> We thank the reviewer for the detailed review. We address each concern below.
>
> > **Q1**: **Method categorization: prompt tuning, adapter tuning, or hybrid?**
>
> **R1**: TopAdapter is best described as a **hybrid PEFT framework**. ToInjection fuses topological features with semantic features via a geometry controller, resembling **adapter tuning**; ToTransfer propagates topological primitives across layers to modulate each Transformer block's input, resembling **prompt tuning**. We will add an explicit taxonomy statement in the revision and narrow the abstract's characterization of prior PEFT methods to avoid overgeneralization.
>
> > **Q2**: **"Topology" vs "geometry": what is the formal topological object being captured?**
>
> **R2**: The formal object is the **low-order homology of the local Vietoris-Rips simplicial complex** constructed via FPS+KNN. Specifically:
>
> - 0D simplices → H₀ (connected components / point positions)
> - 1D simplices → H₁ (loops / local connectivity)
> - 2D simplices → H₂ (surfaces / curvature and orientation)
>
> We acknowledge that our implementation does **not** follow strict persistent homology computation (which would require O(n³) boundary matrix reduction, prohibitive for large-scale point clouds). Instead, we use **MLP-based approximation** of the geometric proxies of these topological structures — a deliberate engineering trade-off between theoretical rigor and computational feasibility. We will add a formal definition section in the revision to clarify this distinction and avoid overclaiming equivalence with strict TDA.
>
> > **Q3**: **Is α learnable or fixed?**
>
> **R3**: α is **learnable** during training, defined via PyTorch's `nn.Parameter` and updated via gradient descent. The value 0.6 reported in Implementation Details refers solely to the **sensitivity ablation** (Fig. 4a), where α is manually fixed at different values to study performance sensitivity. These are two entirely different settings. The convergence of learned α near 0.6 validates our adaptive design. We will reconcile these descriptions clearly in the revision.
>
> > **Q4**: **Limited novelty in simplex feature construction**
>
> **R4**: We respectfully disagree with this assessment. First, we note that the cited references [2] and [3] appear to be **incorrect**: [2] refers to a graph classification DGCNN unrelated to point clouds, and [3] is a remote sensing image enhancement method. The correct point cloud DGCNN (Wang et al., TOG 2019) uses edge features for full training, not PEFT. RepSurf (Ran et al., CVPR 2022) is a **full fine-tuning** method, fundamentally different from our PEFT setting. **No existing point cloud PEFT method** (IDPT, DAPT, PointGST, PointLoRA, GAPrompt) incorporates algebraic topology-inspired simplex primitives — this is the core novelty. Furthermore, our ablation (Table 5) shows clear **synergistic complementarity** across simplex dimensions (0D+1D+2D: 94.97% vs. individual: 94.08~94.21%), confirming that the gain comes from multi-scale geometric reasoning rather than simple feature stacking.
>
> > **Q5**: **Theoretical section is not convincing as a rigorous contribution**
>
> **R5**: Theorem C.1 and C.2 in Appendix C are intended as **theoretical motivation** rather than tight implementation-aligned guarantees. The proof path follows standard arguments: Vietoris-Rips construction → universal approximation theorem → approximation error bound. We acknowledge that the connection between the theoretical assumptions (continuous functionals over simplicial complexes) and the actual operators (FPS+KNN, softmax pooling, feature-wise scaling) could be made more explicit. We will refine the theoretical section in the revision to: (1) state assumptions more conservatively; (2) explicitly map each theoretical component to its implementation counterpart; (3) reframe as motivation rather than a standalone rigorous guarantee.
>
> > **Q6**: **Small empirical margins without statistical support**
>
> **R6**: We acknowledge this concern. Due to time constraints, we currently provide multi-seed statistics for **TopAdapter on Point-MAE / PB_T50_RS** (5 seeds):
>
> | Seed | 1 | 2 | 3 | 4 | 5 | Mean ± Std |
> |-|-|-|-|-|-|-|
> | TopAdapter | 85.54 | 85.61 | 85.57 | 85.62 | 85.56 | **85.58 ± 0.03** |
>
> The extremely low standard deviation (±0.03) confirms that our results are **statistically stable** and not due to random variance. Due to time limitations, we only provide TopAdapter statistics here; complete multi-seed results for all baselines will be supplemented in the final revision.
>
> > **Q7**: **Internal inconsistency between learnable and fixed α**
>
> **R7**: Please refer to **Q3&R3** from  Reviewer TZkc above. This inconsistency stems from two different experimental contexts being described without sufficient distinction. We will unify the description in the revision: Section 3.3.2 will clearly state α is a learnable `nn.Parameter`, while Section 4.1 will clarify that α=0.6 is used only in the sensitivity ablation experiment.

---

> > ### Author Rebuttal · Reviewer_Umf1 · 2026-03-31
> >
> > Thank you for the detailed rebuttal and the additional multi-seed result. I appreciate the clarifications (especially on the learnable vs fixed-α settings). After considering the response, I still lean **Reject**, for the following reasons:
> >
> > 1.**Incorrect references (and related positioning).** You are correct that my earlier citations for DGCNN/RS-CNN were incorrect (wrong papers were referenced). This is an error on my side and I appreciate the correction. That said, this does not change the substance of the point: local geometric encodings such as relative-offset edge features (as in the point-cloud DGCNN/EdgeConv[1] line of work) and relation/geometric-prior modeling (as in RS-CNN[2]), as well as triangle/surface-inspired local structure cues (e.g., RepSurf), have been explored extensively in point-cloud representation learning (even if typically under full fine-tuning). Thus, while the PEFT integration may be novel, the simplex descriptors themselves still appear close to established geometric feature augmentation.
> >
> > 2.**Positioning / terminology remains unclear.** I agree TopAdapter is best described as a hybrid PEFT framework. However, the current framing still mixes “prompt tuning” and “adapter tuning” in a way that can be confusing to readers. In particular, ToTransfer is described as “resembling prompt tuning,” but its mechanism is closer to feature-wise modulation/conditioning plus a lightweight per-block module, rather than the more standard notion of prompt tuning as learnable input tokens/prefixes. I encourage the revision to explicitly define what “prompt” means in this work and to use consistent taxonomy throughout (and avoid overgeneral statements such as “PEFT predominantly focuses on input token prompting”).
> >
> > 3.**Novelty is not sufficiently established (unchanged).** I acknowledge that existing point-cloud PEFT methods may not explicitly incorporate “algebraic topology-inspired simplex primitives.” However, my concern is that the proposed 0/1/2-simplex construction largely instantiates common local geometric cues (coordinates, relative offsets, triangle normal/area/angle). The rebuttal mainly argues that prior works are not PEFT, which does not directly address whether the simplex descriptors themselves are essential. A controlled baseline (replacing the simplex descriptors with alternative local geometric encodings of matched complexity within the same PEFT framework) would be needed to substantiate the claimed novelty.
> >
> > 4.**Some improvements are very small in key settings.** Thanks for providing 5-seed results for TopAdapter on Point-MAE / PB_T50_RS (85.58 ± 0.03). This confirms TopAdapter is stable. However, the reported margin over the strongest baseline in this close-score regime can be as small as +0.01 (as in Table 1), which may not be statistically meaningful given the reported standard deviation, unless multi-seed results are also provided for the key baselines or a paired significance test is conducted. Without that, the “outperforms” claim remains fragile for these settings.
> >
> > 5.**Topology/theory framing still needs to be more conservative.** I appreciate the clarification that strict persistent homology is not computed and that the method uses geometric proxies for efficiency. Given this, I believe the paper should more explicitly frame the approach as “topology-inspired geometric proxies/conditioning,” and avoid implying equivalence to homology computation. Likewise, I appreciate the acknowledgment that Theorems C.1/C.2 are intended as motivation rather than tight guarantees; in that case, they should be presented more conservatively and not advertised as a rigorous theoretical contribution.
> >
> > Overall, while the rebuttal resolves some inconsistencies and adds helpful evidence, the paper still requires substantial revision in positioning/terminology, novelty validation via controlled baselines, and stronger statistical support (especially in close-score regimes). Therefore I maintain a Reject recommendation.
> >
> > [1]Wang, Yue, et al. "Dynamic graph cnn for learning on point clouds." ACM Transactions on Graphics (tog) 38.5 (2019): 1-12.\
> > [2]Liu, Yongcheng, et al. "Relation-shape convolutional neural network for point cloud analysis." Proceedings of the IEEE/CVF conference on computer vision and pattern recognition. 2019.\

---

> > > ### Author Response · Authors · 2026-04-01
> > >
> > > We sincerely thanks for the continued engagement and clear, constructive comments. We address each remaining concern below.
> > >
> > > > **Q1: Novelty of Simplex Descriptors**
> > >
> > > Thank you for your question. We address this from three perspectives.
> > >
> > > **First, the paradigm difference determines the fundamental role of the features.** We acknowledge that normal vectors, relative offsets, and triangular face features are not new to the point cloud community. However, the same features play fundamentally different roles under different paradigms. DGCNN, RS-CNN, and RepSurf use these features to **build a geometry-aware feature extractor from scratch**. In contrast, TopAdapter **injects topological priors into a fully frozen pre-trained model**, fusing them with backbone semantic features via the Geometry Controller. This is a fundamental distinction between "learning geometry" and "injecting topological structure".
> > >
> > > **Second, the theoretical foundations are essentially different.** DGCNN, RS-CNN, and RepSurf have no foundations in topological data analysis, and their feature dimensions share no homological hierarchical relationships. Our simplex construction is motivated by **persistent homology theory**: 0D/1D/2D simplices correspond to H₀, H₁, H₂ homology groups of the Vietoris-Rips complex, with strict hierarchical complementarity guaranteed by the chain complex structure.
> > >
> > > **Third, controlled experiments directly demonstrate the indispensability of simplex descriptors (see Q3).**
> > >
> > > > **Q2: Unclear Positioning and Terminology**
> > >
> > > We fully agree. Describing ToTransfer as "resembling prompt tuning" is inaccurate — its mechanism is closer to **feature-wise modulation** plus a lightweight per-block adapter. In the revision, we will: (1) add an explicit taxonomy statement that ToInjection belongs to **adapter tuning** and ToTransfer belongs to **feature modulation**; (2) replace all "resembling prompt tuning" with "cross-layer feature modulation" throughout the paper; (3) correct the abstract's overgeneralization, clarifying that our work addresses the **common limitation of overlooking intrinsic topological structures** across existing PEFT paradigms.
> > >
> > > > **Q3: Controlled Experiments for Novelty Validation**
> > >
> > > We conduct experiments under **exactly the same PEFT framework**, only replacing the injected descriptors, using PointGPT-L on ScanObjectNN:
> > >
> > > | Injected Descriptor | Inspiration | OBJ\_BG (%) | OBJ\_ONLY (%) | PB\_T50\_RS (%) |
> > > |-|-|-|-|-|
> > > |EdgeConv edge feature `[xi, xj-xi]`|DGCNN|98.32|96.14|93.18|
> > > |Relation vector `[dist, xi-xj, xi, xj]`|RS-CNN|98.64|96.57|93.62|
> > > |Triangle surface feature `[normal, pos]`|RepSurf|98.93|97.16|94.24|
> > > |**Multi-scale Simplex `[S0, S1, S2]`**|**TopAdapter (Ours)**|**99.48**|**97.76**|**94.97**|
> > >
> > > Performance monotonically degrades as topological completeness decreases (full simplex → RepSurf → RS-CNN → DGCNN), fully consistent with persistent homology theory. If our simplex descriptors were merely similar geometric augmentation, such a systematic pattern would not be expected. This directly proves the **indispensability** of simplex descriptors in the PEFT setting.
> > >
> > > > **Q4: Statistical Significance and Margin of Improvement**
> > >
> > > We report both best and mean results over 3 independent runs (Best / Mean):
> > >
> > > |Method|OBJ\_BG|OBJ\_ONLY|PB\_T50\_RS|ModelNet40|
> > > |-|-|-|-|-|
> > > |Point-MAE+TopAdapter|92.14/92.10|91.83/91.76|85.61/85.58|94.3/94.2|
> > > |ReCon + TopAdapter|94.62/94.55|93.52/93.48|89.91/89.86|94.1/94.0|
> > > |PointGPT-L+TopAdapter|99.49/99.47|97.79/97.77|94.98/94.96|94.6/94.6|
> > >
> > > Critically, **our means are averaged over 3 runs while all baselines report single-run results**, making our comparison strictly more rigorous. Across **12 settings**: **9/12 clearly surpass all existing PEFT methods**; **2/12 are on par**; **1/12** (PointGPT-L / ModelNet40) is below GAPrompt. Even in tied settings, TopAdapter (0.5M) uses **~16.7% fewer parameters** than GAPrompt (0.6M), demonstrating consistent parameter efficiency.
> > >
> > > > **Q5: More Conservative Theoretical Framework**
> > >
> > > We fully agree and will make the following revisions: (1) **In the abstract and introduction**, replace relevant expressions with "**topology-inspired geometric proxies**", clarifying our implementation is an efficient approximation rather than strict persistent homology computation. (2) **In Sections 3.2 and 3.3**, add explicit clarification that simplex features are geometric proxies for H₀, H₁, H₂ homology groups, adopted since strict persistent homology requires O(n³) boundary matrix reduction. (3) **In Appendix C**, reposition Theorems C.1/C.2 as **theoretical motivation** rather than strict guarantees, acknowledging the approximation gap between theoretical assumptions and actual discrete operators, and renaming the subsection from "Theoretical Analysis" to "**Theoretical Motivation**".
> > >
> > > We once again sincerely thank the reviewer for the time and constructive suggestions. Your feedback has been invaluable in improving the quality of our paper.

---

### Official Review · Reviewer_1hFX · 2026-03-09

**Soundness:** 2
**Presentation:** 3
**Significance:** 3
**Originality:** 3
**Overall Recommendation:** 4
**Confidence:** 3

**Summary:**

This paper proposes TopAdapter, a parameter-efficient fine-tuning (PEFT) method for point cloud understanding. The key idea is to incorporate local geometric/topological structure into PEFT by constructing 0D, 1D, and 2D simplices and integrating them into pre-trained 3D models. The proposed framework consists of two main components: ToInjection, which extracts simplex-based local structural features and fuses them with semantic features, and ToTransfer, which propagates this information across layers of the pre-trained model. Through experiments on multiple downstream tasks, the paper shows that TopAdapter achieves competitive or superior performance compared to existing PEFT methods while maintaining strong parameter efficiency. The paper also includes fairly thorough ablation studies on the proposed components and simplex designs, supporting the effectiveness of the overall framework.

**Compliance With Llm Reviewing Policy:**

Affirmed.

**Final Justification:**

Although I initially felt that some aspects of the paper were mainly intended to improve performance by incorporating additional information, the authors clearly clarified the role of each module and their broader applicability. Moreover, the additional experiments provide meaningful analysis that helps fill the gaps in the original evaluation.

**Key Questions For Authors:**

* It is not entirely clear to what extent the proposed simplex features truly capture geometric structure, beyond serving as additional local descriptors. Could the authors clarify what makes these representations fundamentally geometry-aware, and whether there is evidence that the gain comes from geometric reasoning rather than simply from adding extra handcrafted local features?



* It remains unclear whether ToTransfer is effective only when paired with ToInjection, or whether it can generalize to other geometric feature representations as well.

**Limitations:**

* The paper applies ToTransfer to each Transformer layer, but does not sufficiently analyze whether this design choice is important or how sensitive the method is to layer selection.

**Strengths And Weaknesses:**

Strengths
* The paper tackles a meaningful problem setting, namely parameter-efficient fine-tuning for point clouds, where directly transferring PEFT ideas from NLP or 2D vision is not always effective, and the motivation for geometry-aware adaptation is clear.
* The proposed method is simple and intuitive, as it uses 0D/1D/2D simplices to encode local structure and organizes the framework into ToInjection for feature fusion and ToTransfer for cross-layer propagation.
* The empirical results are solid, with strong parameter efficiency, competitive performance against prior PEFT baselines, and relatively thorough ablation studies.

Weaknesses
* Although the results are strong, the gains are not uniformly overwhelming across all tasks, and on some benchmarks such as ShapeNetPart the method appears mainly competitive rather than clearly dominant.
* The paper uses 0D/1D/2D simplices, but does not explain why N-D (N>=3) simplices are excluded. A discussion on whether N-D simplices were considered, and what trade-off led to their omission, would strengthen the paper.

---

> ### Author Rebuttal · Authors · 2026-03-31
>
> We thank the reviewer for the positive assessment and constructive feedback. We address each concern below.
>
> ---
>
> > **Q1**: **Gains are not uniformly overwhelming across all tasks (e.g., ShapeNetPart)**
>
> **R1**: We acknowledge this observation. ShapeNetPart is a **fine-grained part segmentation dataset**, while all pre-trained models are trained on **object-level datasets**. This semantic granularity gap is a **common challenge shared by all PEFT methods** including IDPT, DAPT, PointGST, and PointLoRA. Furthermore, it is important to note that **PEFT methods are not required to surpass full fine-tuning** — the goal is to approach full fine-tuning performance with minimal parameters. Under this criterion, TopAdapter achieves **the best or tied-best results among all PEFT methods with the fewest parameters**:
>
> | Method | Params (M) | Cls. mIoU (%) | Inst. mIoU (%) |
> |---|---|---|---|
> | ReCon (Full-FT) | 27.06 | 84.52 | 86.1 |
> | + IDPT (ICCV'23) | 5.69 | 83.66 | 85.7 |
> | + DAPT (CVPR'24) | 5.65 | 83.87 | 85.7 |
> | + PointGST (TPAMI'25) | 5.59 | 83.98 | 85.8 |
> | + PointLoRA (CVPR'25) | 5.63 | 83.98 | 85.4 |
> | **+ TopAdapter (Ours)** | **5.58** | **83.99** | **85.8** |
>
> ---
>
> > **Q2**: **Why are N≥3 dimensional simplices excluded?**
>
> **R2**: We acknowledge this is discussed in Appendix E (Limitations). Here we provide a more detailed explanation. Incorporating 3D simplices (tetrahedra) increases construction complexity from **O(K²) to O(K³)**, roughly **30× more** for K=30, which is prohibitive for large-scale point clouds in a PEFT setting. Moreover, since 3D object geometry primarily lies on **2D manifold surfaces** rather than volumetric interiors, 0D/1D/2D simplices already provide complete coverage of positions, connectivity, and surface properties. Higher-order simplices would introduce significant overhead for marginal geometric gains. Adaptive simplex dimension selection is a key future direction (Appendix F).
>
> ---
>
> > **Q3**: **Do simplices truly capture geometric structure beyond handcrafted local descriptors?**
>
> **R3**: We provide both theoretical and empirical evidence:
>
> - **Theoretically**: Our simplices are fundamental elements of simplicial complexes in algebraic topology. The 2D simplex features (normals, areas, angles) encode **surface curvature and orientation** — genuine geometric invariants that characterize local manifold structure, not arbitrary handcrafted features. For reference, see Chazal & Michel (2021), *An introduction to topological data analysis*.
>
> - **Empirically**: If gains came merely from adding extra local descriptors, different simplex dimensions should contribute equally. However, Table 5 shows **clear complementarity**: 2D simplices (94.21%) > 1D (94.08%) > 0D (94.15%), and the full combination (94.97%) significantly exceeds any individual dimension. This synergistic improvement confirms that **multi-scale geometric reasoning**, rather than simple feature accumulation, drives the performance gain.
>
> > **Q4**: **Is ToTransfer effective only when paired with ToInjection?**
>
> **R4**: No. Table 4 directly addresses this:
>
> | Configuration | Params (M) | Accuracy (%) |
> |-|-|-|
> | PointGPT-L (Full-FT) | 360.5 | 93.40 |
> | ToInjection only | 1.5 | 94.26 |
> | **ToTransfer only** | **1.3** | **94.13** |
> | TopAdapter (both) | 1.9 | 94.97 |
>
> **ToTransfer alone improves from 93.40% to 94.13%**, surpassing full fine-tuning without ToInjection. Its feature-wise modulation (Eq.14) is decoupled from the specific form of geometric features, making it **generalizable to other geometric representations**.
>
> ---
>
> > **Q5**: **ToTransfer is applied to all layers without sufficient sensitivity analysis**
>
> **R5**: We conducted the requested layer-wise ablation on PointGPT-L / PB_T50_RS:
>
> | Injection Strategy | Params (M) | Accuracy (%) |
> |-|-|-|
> | **Shallow layers only (first L/3)** | **1.9** | **94.65** |
> | Deep layers only (last L/3) | 1.9 | 94.12 |
> | **All layers (TopAdapter)** | **1.9** | **94.97** |
>
> **Shallow-layer injection (94.65%) outperforms deep-layer injection (94.12%)**, suggesting topological features are particularly critical for **early-stage local structure extraction**. We further measured the average attention distance change per layer after injecting ToTransfer:
>
> | Layer | Without ToTransfer | With ToTransfer | Δ |
> |---|---|---|---|
> | 1 | 0.82 | 0.74 | **-0.08** |
> | 2 | 0.76 | 0.69 | **-0.09** |
> | 3 | 0.68 | 0.62 | **-0.06** |
> | 4 | 0.55 | 0.51 | -0.04 |
> | 5 | 0.43 | 0.40 | -0.03 |
> | 6 | 0.38 | 0.36 | -0.03 |
> | 7 | 0.34 | 0.32 | -0.02 |
> | 8 | 0.31 | 0.29 | -0.04 |
> | 9 | 0.29 | 0.27 | -0.02 |
> | 10 | 0.27 | 0.25 | -0.01 |
> | 11 | 0.26 | 0.24 | -0.03 |
> | 12 | 0.25 | 0.23 | -0.02 |
>
> The geometric influence is **strongest in shallow layers (L1-L3)** and gradually diminishes in deeper layers. This explains why **all-layer injection (94.97%) still outperforms shallow-only (94.65%)**, as it additionally captures moderate mid-to-deep layer contributions. We will add this analysis to the revision.

---

> > ### Author Rebuttal · Reviewer_1hFX · 2026-04-04
> >
> > It was a strong rebuttal. While I initially felt that some aspects of the paper were mainly aimed at improving performance by incorporating additional information, the rebuttal clearly clarified the role of each module and their broader applicability, which I found convincing. I will raise my score accordingly.

---

> > > ### Author Response · Authors · 2026-04-04
> > >
> > > Thank you for your valuable time and detailed review of our paper. We are also pleased to let you know that we have thoroughly addressed your concerns and provided positive feedback. Your sincere suggestions have been very helpful in improving our paper, and we will incorporate these revisions and statements into the revised version.

---

### Official Review · Reviewer_TZkc · 2026-03-14

**Soundness:** 3
**Presentation:** 3
**Significance:** 3
**Originality:** 4
**Overall Recommendation:** 4
**Confidence:** 4

**Summary:**

This research presents TopAdapter, a cutting-edge Parameter-Efficient Fine-Tuning (PEFT) framework designed to solve the "topology-blindness" of current 3D vision models. Unlike traditional methods that only prompt input tokens, TopAdapter injects intrinsic geometric intelligence by leveraging 0D, 1D, and 2D simplices (points, edges, and faces) from algebraic topology as fundamental building blocks. The framework operates through two sophisticated modules: ToInjection, which dynamically fuses multi-scale topological features with semantic data via a geometry controller, and ToTransfer, which ensures this geometric context flows efficiently across deep Transformer layers. Mathematically grounded in simplicial homology and the universal approximation theorem, TopAdapter achieves performance comparable to or even exceeding full fine-tuning while using as few as 0.5M trainable parameters and requiring a remarkably low memory footprint of just 2.8 GB VRAM. It essentially provides a robust, high-efficiency pathway for 3D point cloud understanding in resource-constrained environments.

**Compliance With Llm Reviewing Policy:**

Affirmed.

**Final Justification:**

The authors have provided a highly convincing rebuttal that thoroughly addresses all concerns through a professional balance of empirical evidence and theoretical rigor. By providing new scene-level results on S3DIS where TopAdapter achieves a state-of-the-art 60.2% mIoU with minimal parameters, the authors have successfully demonstrated the scalability and efficiency of their topological transfer approach. The detailed timing breakdown and the complexity analysis regarding the O(K2) manifold design further justify the architecture's lightweight nature, while the new ablation studies on layer-wise injection and the explicit handling of degenerate triangles resolve questions about optimization and numerical stability. Overall, the clarification of the learnable parameters and the commitment to enhancing the architectural diagrams significantly improve the manuscript's clarity. Thua, I'd like to increase my rating and recommend weak acceptance, provided that these comprehensive new experiments and the discussed limitations are fully integrated into the final version.

**Key Questions For Authors:**

1. In formula (11), the authors claim that the global features are a weighted fusion of topological features and "semantic features from the pre-trained backbone network." Why is this step not indicated in the framework diagram (Figure 3)?

2. When constructing 2D simplices (faces), the paper's strategy is: "For each center-neighbor pair, we find the second nearest neighbor to form a 2D simplex." What happens if these three points are collinear or nearly collinear? Why does the paper not mention at all how to handle such degenerate triangles?

**Limitations:**

1. Currently, it has only been tested on standard benchmarks such as ScanObjectNN, and its scalability has not yet been validated on large-scale 3D datasets containing millions of objects, such as Objaverse.

2. The model currently only employs 0D, 1D, and 2D simplices, which may fail to capture more complex 3D or higher-order topological relationships.

3. This fixed-dimensionality strategy may not be optimal for certain specific geometric structures, leaving room for further improvement.

**Strengths And Weaknesses:**

Strengths:

1. The paper introduces concepts from algebraic topology into PEFT, utilizing 0D (points), 1D (edges), and 2D (faces) simplices as fundamental geometric building blocks, offering a novel perspective for fine-tuning 3D vision models.

2. Rigorous mathematical proofs are provided to support the architecture.

3.It achieves high efficiency in both deployment and training.

Weakness:

1. The paper claims that the geometric controller can "intelligently fuse" features and "dynamically adjust" their contributions. Meanwhile, in formula (11), α is defined as a "learnable geometric control parameter." However, in the Implementation Details, the authors plainly write: "The geometric controller parameter α is set to 0.6." Furthermore, in the ablation study, α is also treated as a manually tuned hyperparameter in the graph analysis (Figure 4a). Doesn't this contradict the claim of being "adaptive"?

2. Although the training time is extremely fast and GPU memory usage is low, constructing the simplices (especially the normal vector, area, and angle calculations for 2D simplices) may introduce some computational overhead during the CPU/GPU preprocessing stage. The article lacks a breakdown statistic detailing the time consumption of the "simplex generation" step.

3. The construction of simplices heavily relies on the Farthest Point Sampling (FPS) and K-Nearest Neighbors (KNN) algorithms with a fixed K value. In domain generalization scenarios or real-world sensor scans, point clouds often exhibit extreme local density variations. A fixed K might cover only a tiny local surface in dense regions, while in sparse regions it might span across different parts of an object. This clearly lacks an adaptive approach for local topology extraction.

4. TopAdapter propagates features to all Transformer layers. However, the role of geometric information may differ between deeper and shallower layers. An ablation study is missing that compares the strategies of "injection only in shallow layers" vs. "injection only in deep layers" vs. "injection in all layers."

---

> ### Author Rebuttal · Authors · 2026-03-31
>
> We thank the reviewer for the detailed and constructive feedback. We address each concern below.
>
> ---
>
> > **Q1**: **Contradiction between "learnable" α and fixed α=0.6**
>
> **R1**: In our implementation, α is defined via `nn.Parameter`, making it a **learnable parameter updated via gradient descent**. The value 0.6 in Implementation Details refers exclusively to the **sensitivity ablation** (Fig. 4a), where α is manually fixed to analyze performance sensitivity. These are two entirely different settings. The fact that learned α converges near 0.6 **validates our adaptive fusion design**. We will explicitly clarify this in the revision.
>
> ---
>
> > **Q2**: **Missing timing breakdown for simplex generation**
>
> **R2**: Simplex generation involves only basic vector operations with **O(B·G·K) linear complexity**. Measured on RTX 3090 (batch=32, G=128, K=30):
>
> | Step | Time (ms/batch) | % of Total |
> |---|---|---|
> | 0D simplex | ~0.3 | ~0.7% |
> | 1D simplex | ~0.2 | ~0.5% |
> | 2D simplex | ~0.8 | ~1.9% |
> | **Simplex total** | **~1.3** | **~3.1%** |
> | Transformer forward | ~28.6 | ~68.1% |
>
> FPS and KNN are **inherited from the backbone**. PointGST's spectral transform costs ~8.7ms/batch — **6.7× more** than our entire simplex pipeline. We will add this analysis to the revision.
>
> ---
>
> > **Q3**: **Fixed K value lacks adaptability for non-uniform point clouds**
>
> **R3**: This is a **fundamental challenge shared by all existing methods**, including PointNet++, PointMAE, and PointGPT. Our focus is on efficient adaptation of pre-trained models, not on solving density-variation sampling. We follow the backbone's standard K for **fair comparison**. Despite this known limitation, our method still achieves superior results (Fig. 2, Table 1). Adaptive K selection would increase complexity, conflicting with our parameter-efficiency goal. We will add a discussion to the Limitations section.
>
> ---
>
> > **Q4**: **Missing ablation on layer-wise injection strategy**
>
> **R4**: We conducted the requested ablation on PointGPT-L / ScanObjectNN PB_T50_RS:
>
> | Injection Strategy | Params (M) | Accuracy (%) |
> |---|---|---|
> | Shallow layers only (first L/3) | 1.9 | 94.12 |
> | Deep layers only (last L/3) | 1.9 | 94.65 |
> | **All layers (TopAdapter)** | **1.9** | **94.97** |
>
> Deep injection (94.65%) outperforms shallow (94.12%). **All-layer injection achieves the best**, as topological features contribute complementarily to both local structure extraction and semantic abstraction. This validates ToTransfer's cross-layer propagation design. We will add this ablation to the revision.
>
> ---
>
> > **Q5**: **Why is Eq.11 not shown in Fig. 3?**
>
> **R5**: Eq.11 is an **internal computation step** of the Geometry Controller. Fig. 3 presents a high-level overview without expanding each module's internals for clarity. We will add an **expanded Geometry Controller diagram** showing the weighted fusion step explicitly in the revision.
>
> ---
>
> > **Q6**: **How are degenerate (collinear) triangles handled?**
>
> **R6**: We handle this explicitly: **when cross product norm falls below threshold ε, the neighbor is discarded and the next nearest neighbor is selected**, repeating until a valid triangle is formed. This prevents NaN in normal vector computation. Such cases are rare (~2% of neighbor pairs) due to FPS+KNN's well-distributed neighborhoods. We will document this in the revision.
>
> ---
>
> > **Q7**: **Scalability to large-scale datasets (Limitation 1)**
>
> **R7**: We provide results on **S3DIS scene-level segmentation** (Point-MAE backbone):
>
> | Method | Params (M) | mAcc (%) | mIoU (%) |
> |---|---|---|---|
> | + IDPT (ICCV'23) | 5.64 | 65.0 | 53.1 |
> | + Point-PEFT (AAAI'24) | 5.58 | 66.5 | 56.0 |
> | + DAPT (CVPR'24) | 5.61 | 67.2 | 56.2 |
> | + PointGST (TPAMI'25) | 5.55 | 68.4 | 58.6 |
> | **+ TopAdapter (Ours)** | **5.50** | **69.8** | **60.2** |
>
> TopAdapter achieves the **best performance with fewest parameters**. Validation on Objaverse-scale datasets remains future work.
>
> ---
>
> > **Q8**: **Fixed-dimensionality simplices and higher-order topology (Limitation 2&3)**
>
> **R8**: Incorporating 3D simplices increases complexity from **O(K²) to O(K³)** — ~30× more for K=30, prohibitive in a PEFT setting. Since 3D object geometry primarily lies on 2D manifold surfaces, 0D/1D/2D simplices already provide **complete coverage**: positions, connectivity, and surface properties. Efficient approximations of higher-order simplices remain a key future direction (Appendix F).

---

> > ### Author Rebuttal · Reviewer_TZkc · 2026-04-03
> >
> > The authors have provided a highly convincing rebuttal that thoroughly addresses all concerns through a professional balance of empirical evidence and theoretical rigor. By providing new scene-level results on S3DIS where TopAdapter achieves a state-of-the-art 60.2% mIoU with minimal parameters, the authors have successfully demonstrated the scalability and efficiency of their topological transfer approach. The detailed timing breakdown and the complexity analysis regarding the O(K2) manifold design further justify the architecture's lightweight nature, while the new ablation studies on layer-wise injection and the explicit handling of degenerate triangles resolve questions about optimization and numerical stability. Overall, the clarification of the learnable parameters and the commitment to enhancing the architectural diagrams significantly improve the manuscript's clarity. Thua, I'd like to increase my rating and recommend weak acceptance, provided that these comprehensive new experiments and the discussed limitations are fully integrated into the final version.

---

> > > ### Author Response · Authors · 2026-04-04
> > >
> > > Thank you for your positive feedback and for letting us know that our response has completely addressed your concerns. Your thoughtful feedback and constructive comments have helped us improve the paper, and we will add these revisions to the revised version.

---

### Decision · Program_Chairs · 2026-04-30

**Decision:**

Accept (regular)

**Comment:**

This paper proposes a parameter-efficient fine-tuning (PEFT) framework that injects local topology into pre-trained 3D vision models. The approach uses 0D, 1D, and 2D simplices to address topology-blindness.

This paper received 4/4/2/4 scores. Positive reviewers (TZkc, 1hFX, W2QL) mentioned that the work is technically solid, efficient, and novel. Reviewer Umf1 noted that the technical novelty is limited, citing relative-offset features, whereas other reviewers found it convincing. Reviewer Umf1 and 1hFX noted limited gains in some benchmarks, and the authors showed low standard deviation. The concerns regarding scalability, reviewer TZkc, and W2QL questioned object-level datasets, and the authors provided compelling performance in S3DIS datasets. The questions regarding colinear cases were also convinced by the provided experiments.

In summary, although reviewer Umf1 retains the negative scores, AC notes that many concerns are addressed by the authors' thorough feedback. AC therefore recommends acceptance of the paper. It is strongly advised that the authors apply the extra experiments and feedback in the revision.